# Multidose Hyaluronidase Administration as an Optimal Procedure to Degrade Resilient Hyaluronic Acid Soft Tissue Fillers

**DOI:** 10.3390/molecules28031003

**Published:** 2023-01-19

**Authors:** Killian Flégeau, Jing Jing, Romain Brusini, Mélanie Gallet, Capucine Moreno, Lee Walker, François Bourdon, Jimmy Faivre

**Affiliations:** 1Research and Development Department, Teoxane SA, Rue de Lyon 105, 1203 Geneva, Switzerland; 2Private Practice, B City Clinic, 88 Rodney Street, Liverpool L1 9AR, UK

**Keywords:** hyaluronic acid, hyaluronidase, soft tissue filler, enzymatic degradation, rheology

## Abstract

Minimally invasive hyaluronan (HA) tissue fillers are routinely employed to provide tissue projection and correct age-related skin depressions. HA fillers can advantageously be degraded by hyaluronidase (HAase) administration in case of adverse events. However, clear guidelines regarding the optimal dosage and mode of administration of HAase are missing, leaving a scientific gap for practitioners in their daily practice. In this study, we implemented a novel rheological procedure to rationally evaluate soft tissue filler degradability and optimize their degradation kinetics. TEOSYAL RHA^®^ filler degradation kinetics in contact with HAase was monitored in real-time by rheological time sweeps. Gels were shown to degrade as a function of enzymatic activity, HA concentration, and BDDE content, with a concomitant loss of their viscoelastic properties. We further demonstrated that repeated administration of small HAase doses improved HA degradation kinetics over large single doses. Mathematical analyses were developed to evaluate the degradation potential of an enzyme. Finally, we tuned the optimal time between injections and number of enzymatic units, maximizing degradation kinetics. In this study, we have established a scientific rationale for the degradation of HA fillers by multidose HAase administration that could serve as a basis for future clinical management of adverse events.

## 1. Introduction

Hyaluronan (HA)-based soft-tissue fillers are class III medical devices designed to be injected into skin layers and fat compartments to restore/create volume or correct defects [1]. HA-based filler composition (molecular weight, concentration, crosslinker content) can be tuned to optimize corrective effects and adapt to specific anatomical locations and requirements.

HA is a ubiquitous linear high molecular weight glycosaminoglycan composed of the repetitive disaccharide unit β-(1,3)-N-acetyl glucosamine and β-(1,4)-D-glucuronic acid. The human body contains approximately 15 g of HA, with nearly 50% of the total body’s HA content found in the skin [2]. The degradation of HA is mostly by endogenous hyaluronidases (HAase), a class of enzyme, clipping the β-(1 → 4) glycosidic bond [3]. Their potency, or enzymatic activity, is determined by turbidimetric assays, which may vary between European and American standards [4]. It is measured in units, defining the amount of enzyme catalyzing the conversion of a substrate per unit of time. HA is also sensitive to other endogenous or exogenous degradation factors, including reactive oxygen species (ROS), temperature, pH, and mechanical stimulation [5]. As a result, native HA exhibits a short half-life in tissues, from a few hours in blood to 1–2 days in the skin [2]. To increase its residence time within the target tissue, HA chains can be crosslinked to form viscoelastic covalent networks, known to drastically slow down the turnover from a few days to several months [6]. Although numerous functionalization pathways were developed to create HA hydrogels [7,8,9], 1,4 butanediol diglycidyl ether (BDDE) remains the gold standard for soft tissue fillers, including the TEOSYAL RHA^®^ collection (Teoxane SA, Geneva, Switzerland) [10]. In alkaline conditions, BDDE reacts with the hydroxyl moieties of HA to form an elastic polymer network, slowing down the degradation process without hampering gel degradation via β-(1,4) glycosidic bonds cleavage by HAase [3,11,12,13].

Owing to unmatched biocompatibility and bioactivity, the advent of HA hydrogels has heralded a new era in the field of aesthetic medicine [14]. However, with the increasing use of HA fillers comes along a drastic rise in the incidence of soft tissue filler adverse events (e.g., unacceptable cosmetic outcomes, blindness, allergies, hypersensitivity, and vascular-occlusive injuries) [15,16,17]. To overcome these burdensome complications, appropriate injections of exogenous HAase are performed by practitioners to degrade the injected filler. Commercial HAases originate from mammalian, leech, and microbial origins [18] with enzymatic preparations ranging from 150 to 1500 U·mL^−1^ [19]. Although practical guidelines were published to help practitioners in their correction choice [15,20,21,22], they are based on procedures compiled from decades of clinical observations and expert opinion. Since HA degradation strongly depends on the placement and resistance of the gel (HA concentration and crosslinking), the volume and activity of the enzymatic solution [6], an evidence-based procedure to evaluate and accelerate gel degradation would be highly desirable. However, to the best of our knowledge, guidelines based on a rationale for physicochemical evaluation of gel degradation have not yet been published.

Numerous enzymatic degradation tests of HA fillers are reported in the literature, including visual assessment of degradation [23,24,25], in vitro quantification of released HA fragments [12,13,26,27], and palpation or imaging in human patients [28,29] and animal models [30]. This diversity of procedures and a lack of regulation regarding the optimal degradation procedures make the evaluation of HA filler degradability and safety particularly challenging [31]. In addition, none of these tests quantitatively report the physicochemical alteration of HA gel properties during degradation. To overcome these limitations, we established a new real-time monitoring of gel degradation based on rheological time sweeps. This new protocol was used to evaluate the degradation kinetics of the TEOSYAL RHA^®^ collection, manufactured by TEOXANE SA (Geneva, Switzerland) using the preserved network technology^®^ (PNT) [10]. Two enzymes were compared, the Hylenex 150 USP U·mL^−1^ from Halozyme Therapeutics, an FDA-approved enzyme in use in the U.S, and the Hyalase 1500 U.I from Wockhardt, approved in Europe. Gels were shown to lose their elastic properties over time, transitioning to a liquid-like state as a function of the enzyme source, crosslinking density, and HA concentration. To optimize gel degradation kinetics, comparison between single, double, and triple dose injections of an enzyme were performed for PNT4, the stiffest gel of the RHA collection. The multidose technique showed an improved ability to degrade the gels over standard single injection procedures. Different timings of injection and the number of enzymatic activities were further screened in a multidose mode to fine-tune the degradation process. Finally, an optimized degradation protocol was established and tested on all PNT gels to demonstrate the improved ability of the multidose procedure to degrade HA-based hydrogel. We developed, for the first time, a real-time evaluation protocol of gel degradation kinetics and demonstrated the improved gel resorption obtained with the multiple injections protocol. Future work should evaluate the multidose protocol in an in vivo context to further refine enzymatic volumes and units. The aims of this article are to establish new comprehensive guidelines for the degradation of fillers, with the final goal of providing clinicians with an evidence-based approach in their management of HA-related adverse events.

## 2. Results

### 2.1. Development of the Degradation Protocol Assessed by Rheological Time Sweeps

Understanding and characterizing HA-based fillers degradability is key to ascertain safety after gel injection. Unfortunately, a standardized and precise monitoring of gel degradation kinetics in vitro is yet to be defined. To address this shortcoming, we implemented a rheology-based degradation test relying on the real-time monitoring of gel storage moduli (G’) over time (Figure 1A,B). While not fully representative of the gel’s physicochemical properties, the G’ was selected as the key variable due to the direct relationship between gel 3D network persistence and its mechanical properties. While PNT gels storage moduli remained constant over time when mixed with a buffer solution (Appendix A), a progressive loss of the elastic properties was observed after mixing gels with 7.1 U·mL^−1^ of either Hylenex or Hyalase enzymes (Figure 1A,B). The degradation kinetics were inversely proportional to the gel’s initial storage modulus, indicating that all other variables held constant, and stiffer gels have increased resistance to degradation (Table 1). In addition, clear differences in the degradation kinetics were observed between the enzymes. While near-complete degradation of the PNT1, PNT2, PNT3, and PNT4 gels were obtained in less than 180 min with Hylenex (Figure 1A and Appendix A), none of the PNT gels reached complete degradation within the same time with Hyalase (Figure 1B). The time required to lose 50% of the initial storage modulus, namely, the half-life, was next calculated (Figure 1E). With the Hylenex enzyme, 6.8, 12.8, 18.6, and 31.4 min were required to degrade 50% of PNT1, PNT2, PNT3, and PNT4, respectively. Similarly, half-life values with the Hyalase enzyme were determined to be 25.2, 62.1, 100.4, and 176.0 min for the PNT1, PNT2, PNT3, and PNT4, respectively. The enzymatic activity of both enzymes was further evaluated by calculating the degradation rates obtained from the first derivatives of the degradation curves (Figure 1C,D). Confirming the first observations, the degradation rates of PNT gels were largely superior with the Hylenex enzymes than those treated with Hyalase (Appendix A). Maximal degradation rates reached 6 to 9.5 Pa·s^−^^1^ with the Hylenex enzyme compared to 2.2 to 3.8 Pa·s^−^^1^ for the Hyalase. In addition, more prolonged degradation activity was observed with the Hylenex enzyme. While Hyalase maintained a degradation rate above 1 Pa·s^−^^1^ (used as standard reference of the “degradation activity”) for 44.7 ± 0.0 min for PNT4 gels, Hylenex could maintain similar degrade rates for more than 79.4 ± 9.0 min (Figure 1F), ultimately leading to an almost complete liquefaction of PNT gels (Figure 1G). With Hylenex, 90% loss of the initial storage modulus (G’_0_) was obtained after 32.4 ± 1.2, 64.6 ± 4.1, 85.3 ± 6.6, and 148.0 ± 1.0 min, for the PNT1, PNT2, PNT3, and PNT4 gels, respectively (Appendix A). Conversely, only one replicate of the PNT1 gel reached 90% degradation within 180 min using the Hyalase enzyme. Hence, although similar enzymatic units were used, drastically different degradation profiles were obtained between the two enzymes. In this first section, we developed a simple and precise rheology-based protocol to study gel degradation over time.

### 2.2. Degradation of PNT4 Gels with Single, Double, or Triple Doses of Hylenex and Hyalase Enzymes

As the enzymatic digestion rate of PNT gels progressively slows down until becoming negligible, we next hypothesized that repeat injections of a fresh enzyme solution could rejuvenate the enzymatic activity and accelerate gel degradation. To evaluate this hypothesis, we compared the degradation rate of PNT4 gels with a single, double, or triple dose of Hylenex or Hyalase using the newly developed rheological assay (Figure 2A). To compensate for the observed differences between the two enzymes, enzymatic units were increased to 34.6 U and 277.8 U of Hylenex and Hyalase per mL of gel respectively. The total volume (90 μL for Hylenex and 60 μL for Hyalase) and total enzymatic activity were matched between single and multidose approaches to guarantee comparable data interpretation. As previously observed, single dose injection of Hylenex or Hylase resulted in a progressive loss of G’ over time (Figure 2B,D). Dividing the enzyme volume into two or three injections initially slowed down gel degradation, both due to a reduced enzymatic volume and activity used per injection. Notwithstanding, fresh injections of HAase rejunevated the enzymatic activity, ultimately outperforming single dose injections and extending gel degradation after 120 min. With Hylenex, double and triple doses accelerated gel degradation, crossing the single dose curve after 28.3 ± 2.2 and 55.2 ± 4.8 min, respectively, and reaching 92.0 ± 3.6 and 98.8 ± 1.3% gel degradation after 120 min, compared to 90.3 ± 1.8% with the single dose (Figure 2F,G). Increasing the number of injections from 2 to 3 did not accelerate gel degradation kinetics but improved the extent of gel degradation after 2 h. Clear differences were also obtained with the Hyalase enzyme, with double and triple doses crossing the single dose curve in 33.4 ± 6.2 and 46.2 ± 0.1 min, respectively, reaching 83.3 ± 9.0 and 90.7 ± 7.4% gel degradation after 120 min, compared to 77.2 ± 5.3% with the single dose. For both enzymes, the enzymatic degradation rates for a single dose was attained a maximum of 5 min after enzyme addition then progressively decreased, becoming lower than 1 Pa·s^−1^ after 60 min and becoming negligible after 2 h (Figure 2C,E). Splitting the enzyme volume into two or three doses renewed the degradation kinetics and induced a strong boost in the degradation rate after each enzyme addition. This boost was proportional to the amount of enzyme added. Irrespective of the timing, volume and enzymatic activity, the degradation rate was maximal for the first 5 min following enzyme addition. The area under the curve (AUC) was next calculated as a measure of the total gel exposure to enzymatic degradation (Figure 2H). For both enzymes, performing two or three injections increased the AUC compared to a single dose. With Hylenex, the AUC increased from 222.3 ± 3.4 to 231.7 ± 1.9 and 255.0 ± 2.0 Pa for the single, double, and triple doses, respectively. Similarly, the AUC increased from 179.6 ± 4.0 to 237.8 ± 5.2 and 289.0 ± 10.0 Pa for the single, double, and triple dose with Hyalase. These results indicate that performing multidose injections increases gel suscceptitibility to enzymatic digestion, irrespective of the enzyme source.

### 2.3. Refining Multidose Parameters: Injection Spans and Enzymatic Units

Building upon the improved gel degradation results obtained with the multidose approach, we further looked for the optimal time between injections and number of enzymatic units. Triple dose degradation tests of PNT4 gels with 34.6 U·mL^−1^ of Hylenex were performed with intervals of 5, 10, 20, or 30 min (Figure 3A), for 120 min in total. As similar enzymatic units were used per injection, the initial slopes were unsurprisingly identical. However, narrowing the time between injections from 30 to 5 min did speed up gel degradation in the early time points. After 30 min, gel degradation reached 66.5 ± 3.2, 66.9 ± 2.0, 51.3 ± 1.3, and 42.4 ± 1.3%, for injection intervals of 5, 10, 20, and 30 min, respectively (Figure 3B). Increasing the time between each interval ultimately increased the extent of gel degradation, reaching 86.9 ± 1.7, 90.7 ± 1.3, 93.8 ± 1.3, and 94.2 ± 0.4% after 120 min, for injection spans of 5, 10, 20, and 30 min, respectively (Figure 3C). Minor differences in the extent of degradation were obtained between 20 and 30 min for the duration of the experiment, suggesting an optimal degradation time around 20 to 30 min. Degradation rates were next calculated and confirmed a maximum degradation rate for the first 5 min after enzyme addition, irrespective of the injection intervals (Appendix A). Correlating with the previous observations, the AUC of the first derivative curves increased with wider inter-injection, demonstrating more efficient gel degradation with longer intervals (Appendix A). Hence, short injection intervals (5 min) maximize the optimal enzymatic degradability over short period of times while longer intervals induced a more efficient degradation over prolonged periods.

PNT4 degradation was next analyzed as a function of the number of enzymatic units. Degradation studies with 9.4, 20.2, 34.6, and 46.2 units of HAase per mL of gel were performed using the triple dose protocol with injections every 20 min (Figure 3D). As expected, increasing the number of units both accelerated and increased gel degradation. After 30 min, increasing the enzymatic units from 9.4 to 46.2 U·mL^−1^ resulted in a gradual increase in gel degradation from 20.7 ± 4.4 to 61.0 ± 3.6% (Figure 3E). After 2 h, gel degradation attained 62.1 ± 9.1, 88.8 ± 4.0, 94.2 ± 0.4, and 95.7 ± 2.4%, using 9.4, 20.2, 34.6, and 46.2 U·mL^−1^, respectively (Figure 3F). Using 9.4 U·mL^−1^ was clearly ineffective in completing gel degradation. Although near 90% gel degradation was obtained using 20.2 U·mL^−1^ HAase, higher HAase doses did not significantly improve the extent of gel degradation.

### 2.4. Optimized Degradation of PNT Gels Using a Multidose Approach

We previously demonstrated that gel degradation kinetics can be accelerated by either increasing the number of enzymatic units or by performing injections every 5 min. As a final step of our optimized degradation protocol, we performed degradation studies for all PNT gels using repeated injections of 38.3 U·mL^−1^ Hylenex every 5 min until reaching 90% degradation and compared the degradation kinetics to a single dose matching enzymatic volumes and activities (Figure 4 and Appendix A). All PNT gels showed faster degradation rates with multidose injections versus single doses, although the differences were maximal for PNT4 and PNT1 gels. PNT4 gels attained 90% degradation in 26.2 ± 0.1 min with 5 doses. With the exact same conditions but with a single dose of Hylenex (191.6 U·mL^−1^), it took 46.9 ± 3.4 min to reach 90% degradation. Similarly, 90% degradation were reached in 25.9 ± 0.4, 26.9 ± 0.3, 19.7 ± 0.8 min for PNT3, PNT2, and PNT1 gels using 5 (PNT2&3) or 4 (PNT1) injections, compared to 31.1 ± 7.3, 29.8 ± 5.8, and 40.9 ± 3.2 min with single doses (Figure 4E,G and Appendix A). Calculating the degradation rates revealed interesting features. Maximum degradation rates (both single and multidose) were proportional to the gels’ MoD or stiffness, reaching −19.8, −16.9, −14.1, and −5.6 Pa·s^−1^, for PNT4, PNT3, PNT2, and PNT1 gels, respectively (Figure 4C,D and Appendix A). With single dose injections, the degradation rate reached a peak 5 min after enzyme addition and progressively decreased to near-zero values after 60 min. Conversely, small injections every 5 min constantly rejuvenated the degradation activity of the enzyme, although a decreased pic intensity was observed over time. The AUCs were systematically higher with the multidose, even though the experiments were left to run for longer periods with the single dose injections. For example, the AUC of PNT4 for the multidose protocol reached 264.0 ± 15.3 Pa versus 228.6 ± 9.2 Pa with a single injection (Figure 4F,H and Appendix A).

We finally monitored the evolution of PNT gel molecular weight over time by HPLC-SEC, following the multidose injection protocol. In line with our rheological data, HPLC-SEC results demonstrated a progressive decrease in HA gel molecular weight, becoming negligible after repeated injections of the enzyme (Appendix A). As previously observed, PNT4 gels showed a more progressive diminution of the molecular weight compared to the other PNT gels. Opposite to the decreased molecular weight, the percentage of low molecular weight HA fragments was shown to progressively increase over time, reaching 85% on average after 70 min. In conclusion, both HPLC and rheology studies showed the rapid degradation of PNT gels following our multidose injection protocol.

## 3. Materials and Methods

### 3.1. Materials

The investigated HA-based soft-tissue fillers were selected from the TEOSYAL RHA^®^ collection and are displayed in Table 1. Hylenex, a human recombinant hyaluronidase (150 USP U·mL^−1^, Halozyme Therapeutics, San Diego, CA, USA) and Hyalase (powder from ovine origin, Hyaluronidase 1500 I.U, Wockhardt, UK) were used for the rheological tests. Hyalase “Dessau” (150 U·mL^−1^, Riemser Arzneimittel AG, Greifswald, Germany) was used for the SEC experiments.

### 3.2. Single Dose Degradation Kinetics of PNT Gels Measured by Rheology

A rheometer (HR-20, TA Instruments, Newcastle, Delaware, USA) was set up with a cone (anodized aluminum, 40 mm, 1°)—plate geometry at 37 °C. A time sweep procedure at an oscillatory strain of 0.1% at 1 Hz was used. At t = 0 s, 50 μL of HAase solution—representing 7.1 U of both Hylenex and Hyalase per mL of gel—was poured in the empty syringe and gently mixed with 1 mL of gel using a Luer-Lock to ensure complete homogenization. Then, 0.3 mL of the mixed solution were deposited onto the Peltier plate and allowed to equilibrate at 37 °C for 5 min before starting the experiment for a duration of 3 h. Experiments with PBS solution instead of the enzyme solution were performed as controls. A solvent trap was used in all experiments to prevent solvent evaporation. First and second derivatives were calculated using the GraphPad Prism 9 software (version 9.3.1, San Diego, CA, USA) using a second order smoothing.

### 3.3. Determination of Half-Lives and Gel Degradation G’0_90%_

The gel degradation kinetics were modeled using a two-phase decay model calculated by the GraphPad Prism 9 software. Best-fit values were used to resolve Equation (1):(1)$$Y_{\left(Half-life\right)}=G^{\prime }plateau+\sigma_{1}e^{-k_{1}t_{1/2}}+\sigma_{2}e^{-k_{2}t_{1/2}}$$

With $G^{\prime }plateau$, $\sigma_{1}$, and $\sigma_{2}$, being constants determined by the fitting model and dependent on the type of gel and enzyme used.

The time required to reach 90% degradation was determined using Equation (2) and resolved using the Excel Solver:(2)$$({G^{\prime }}_{0}-0.9\times {G^{\prime }}_{0})=G^{\prime }plateau+\sigma_{1}e^{-k_{1}t_{90}}+\sigma_{2}e^{-k_{2}t_{90}}$$

### 3.4. Single versus Multi-Dose Degradation Kinetics of PNT4 Gels Measured by Rheology

The protocol for gel degradation kinetics followed by the time sweeps rheology measurement was slightly adapted for the multidose approach. Approximately 300 μL of gel was deposited onto the plate and allowed to equilibrate for 5 min at 37 °C. The upper geometry was subsequently raised, and the enzymatic solution volume was added onto the gel before starting the experiment. This protocol was repeated once, or twice, for the double and triple doses, respectively, within a span of 20 min. For the Hylenex enzyme, 34.6 U·mL^−1^ of gel was used, with enzymatic volumes of 90, 45, or 30 μL corresponding to a single, double, and triple dose. For the Hyalase enzyme, the concentration was increased to 277.8 U·mL^−1^ and the volumes set at 60, 30, and 20 for a single, double, and triple dose, respectively. The duration of the experiment was set to 2 h for each condition and the change in the elastic modulus G’ was monitored over time. A solvent trap was used in all experiments to prevent solvent evaporation. Each experiment was repeated three times. First derivatives and AUC values were calculated using the GraphPad Prism 9 software (version 9.3.1, San Diego, CA, USA) using a second order smoothing.

### 3.5. Refinment Study of Multi-Dose Degradation Kinetics of PNT4 Gels Measured by Rheology

Refinement of gel degradation was performed following a similar protocol to the one used in 5.4. For the injection spans, a triple dose protocol was performed using 34.6 U of Hylenex per mL of gel in total. The interval between the injections was set at 5, 10, 20, or 30 min and the protocol adapted to last for 2 h in total. Similarly, a triple dose protocol was used for the refinement of the number of enzymatic units. An interval time of 20 min between injections was chosen, and the number of enzymatic units was set at 9.4, 20.2, 34.6, and 46.2 units of HAase per mL of gel, respectively. The duration of the experiment was set to 2 h for each condition and the change in the elastic modulus G’ was monitored over time. Each experiment was repeated three times. First derivatives and AUC values were calculated using the GraphPad Prism 9 software (version 9.3.1, San Diego, CA, USA) using a second order smoothing.

### 3.6. Statistical Analysis

Unless otherwise specified, all data are presented as mean ± standard deviation. The statistical analysis was performed in GraphPad Prism 9.4.1 software (681) (GraphPad Software Inc.) except for Figure 1 which did not present statistical data. For all experiments, n ≥ 2, * *p* < 0.05, ** *p* < 0.01, *** *p* < 0.001, **** *p* < 0.0001. The ANOVA mixed-effects model with the Geisser–Greenhouse correction followed by Tukey’s multiple comparisons test were performed in Figure 2. A non-parametric Kruskal–Wallis test followed by Tukey’s multiple comparisons test were used in Figure 3. Non-parametric Kolmogorov–Smirnov tests were used in Figure 4.

## 4. Discussion

Decades of clinical practice have provided practitioners with multiple guidelines on how to use HAase to treat overcorrection or adverse events. While HAase is not labelled for the treatment of adverse events with fillers, it is tolerated as if in the patient’s best interest (2009 guidelines from the United Kingdom’s MHRA) [15]. However, a review of the literature shows that an optimal way to treat complications is yet to be clearly defined [32]. As the soft tissue filler degradation profile is highly dependent on filler composition, crosslinking density, enzymatic activity, anatomical site differences, and even probably patient-specific metabolic profiles [32], providing optimal guidelines to degrade HA fillers is an arduous task that must be tackled with clear recommendations. In this article, we aimed to rationalize clinical guidelines using a real-time monitoring of gel degradation by rheology time sweeps. While providing thoughtful insights on gel degradation behavior, our method could also greatly help manufacturers in their ability to comply with highly demanding regulatory requirements.

We first developed an enzymatic degradation assay based on the real-time evaluation of gel elastic properties by rheology. To the best of our knowledge, no study has ever attempted to analyze gel enzymatic degradation using this approach. Gel degradation kinetics studies with two commercialized HAases, Hyalase and Hylenex, were performed using time-sweep procedures monitoring the storage modulus G’ evolution over time. Low levels of enzymatic units were used to reproduce HAase diffusion and dilution into the surrounding tissues in vivo. Increasing HA concentration from 15 mg·mL^−1^ to 23 mg·mL^−1^ and degrees of modification from 1.9 to 4.1% resulted in slower degradation rates for the PNT gels [10]. These results match data from randomized clinical trials evaluating the durability and clinical performance of TEOSYAL PNT^®^ fillers in vivo [33,34,35,36]. For example, PNT2, PNT3, PNT4 exhibited slower resorption rate than PNT1 after 52 weeks of injections in the nasolabial region. Hence, despite having among the lowest degrees of modification of soft tissue fillers on the market, PNT gels still demonstrated long lasting properties in vivo. Surprisingly, while all PNT gels were sensitive to HAase enzymatic degradation and although an equal number of enzymatic units were used for the degradation tests, very different degradation kinetics were observed, with Hylenex inducing faster gel degradation than Hyalase. Hylenex is a human recombinant enzyme while Hyalase is of ovine origin [22]. Concerns regarding the source of HAase were previously reported [21,27], whereas others showed no significant differences when using enzymes from different origins [25]. Using a human recombinant and ovine HAase, Shumate and co-workers did not observe altered kinetics of degradation with Juvéderm products, but Restylane gels showed different behavior with the two HAases [30]. In addition, methods for determining the enzymatic potency may vary between the American and European Pharmacopeia. A direct comparison between the two enzymatic activities is, therefore, deemed impossible, calling for precautions during clinical practice [30].

Consistent with most reports [37], gel degradation was maximal in the early times of the experiment, then progressively slowed down until becoming negligible. Calculating the instantaneous degradation rates with the first derivatives revealed optimal enzymatic digestion rate for the first 5 min following enzyme addition, matching in vivo reports of HAase turnover in mice approximately 4 min after i.v. injection in plasma and 5 min after subcutaneous injection in the muscle [38]. These observations motivated the development of a multidose approach, based on the hypothesis that smaller but repeated injections of HAase would maximize the degradation potential of the enzymes by rejuvenating their enzymatic activity.

Multidose injections of smaller HAase volumes were compared with a single dose injection of enzyme. PNT4 gel was taken as a model of slowly resorbable soft tissue filler. Matching the empirical data from clinical practice, gel degradation became minimal after 60 min with the single dose, requiring additional doses to complete the enzymatic degradation process [15,21]. Although using smaller amounts of enzyme per injection, the multidose approach extended gel degradation, with a strong boost in the enzymatic digestion activity after each injection. Increasing the injections from 2 to 3 showed improved degradation results, confirming the benefit of repetitive injections. As the enzymatic activity of hyaluronidases was found to progressively slow down over time (without completing gel degradation), we hypothesize that a progressive inactivation of the enzyme occurs, although not yet fully understood in the literature [19]. Hence, performing repetitive fresh enzyme solutions exceeded single injections by renewing the enzymatic digestion process. These results directly match those reported by Lee and co-workers, demonstrating increased rabbit survival rate after vascular occlusion using repeated HAase injections over one large dose of the enzyme [39]. As it is impossible to know how much of the HAase is delivered to the blocked vessel, the multidose approach may also increase the chance of injecting the enzyme in the right location. Repeated injections may also favor HAase passive diffusion across human arteries [40].

Clinicians reporting multidose injections are compelled to use empirical approaches to develop their protocols. To rationalize the multidose concept and help clinicians in their practice, we further analyze the parameters influencing gel enzymatic degradation, hunting for the optimal parameters. Since pH and temperature are not tunable in vivo, we focused on the intervals between enzyme injections and the quantity of enzyme added. As demonstrated in this article and in accordance with experimental studies in humans [20], a trend of a dose–response relationship was found between the amount of enzyme added and the extent of gel degradation. Modest differences in the extent of gel degradation were observed between 34.6 and 46.2 U·mL^−1^ after 120 min, suggesting that the enzyme was almost saturated, with a maximal degradation rate nearly attained. Even though these data can hardly be extrapolated to in vivo conditions, they point out that a smaller but frequently repeated HAase dosage can still be effective in treating adverse events. Reducing HAase dosage might be desirable as the effect of HAase on skin structure remains controversial. Many patients have reported chronic lower eyelid edema after HAase treatment [41], underlying potential adverse esthetic results following HAase treatment. For example, treatment with 150 U·mL^−1^ of hyaluronidase was shown to have little effect on dermal structure in ex vivo peri-ocular skin samples but formed unwanted non-GAG amorphous extracellular deposits [42]. Reducing the amount of HAase via smaller injections might be beneficial in reducing potential adverse effects while staying in the optimal degradation range of the enzyme. However, the dosage used in our in vitro study should be increased for clinical practice, where dilution, enzymatic inhibition, and diffusion of the enzyme occurs.

In practice, the delay between 2 HAase injections is practitioner-dependent, with variations in the application of conservative measures (massage, tapping, heat) and intervals between skin coloration or capillary refill checks [43]. Nevertheless, hourly injections [15] or 15-min spans [39] were previously reported. To evaluate the optimal time between injections maximizing gel degradation, we performed multidose degradation tests with intervals from 5 to 30 min. Two trends emerged based on the extent of gel degradation and degradation rates. Intervals of 5 min showed the fastest gel resorption in early time points, maximizing the optimal enzymatic activity of the enzyme (i.e., 5 min). Conversely, 30-min intervals led to the greatest extent of gel degradation with the minimal amount of enzyme, maximizing the enzyme durability. After 120 min, almost no differences were detected between the 20- and 30-min spans, indicating optimal injections every 20 to 30 min for this specific set-up. Repeated injections with the least amount of time between injections can be suggested to speed up the degradation. If no real emergency is observed, hourly or 30-min injection intervals, as previously mentioned [15], can be largely sufficient to induce gel resorption, confirming previous reports [15]. We concluded this study by performing degradation tests on all PNT gels using an optimized protocol that could be later broadened to in vivo conditions. All PNT gels showed faster degradation with the multidose protocol than with a single dose, although volumes and enzymatic units were matched. Thus, the multidose concept is verified for all BDDE-crosslinked PNT fillers, from very light fillers to volumizers.

Throughout this study, we implemented and demonstrated the benefits of using rheological measurements to model dermal filler enzymatic degradation. The efficacy of the multidose delivery of hyaluronidase in vitro supports the emergency high-dose pulsed (multidose) protocols delivered in vivo by clinicians in accordance with guidelines by DeLorenzi et al. [21]. Future studies will evaluate these experimental protocols in in vivo or ex vivo set up to further validate this concept.

## 5. Conclusions

This study was designed as the first real-time evaluation of HA gel enzymatic degradation evaluated by rheology. This protocol was used to determine the ability of two commercialized enzymes to degrade PNT gels. Surprisingly, clear differences in the degradation ability were observed between Hylenex and Hyalase. We also demonstrated the superiority of the multidose approach over standard single dose injections, with all Teosyal PNT gels showing improved degradation when applying repetitive doses of HAase. Our rationale optimization procedure of gel degradation highlighted three crucial main factors (i) the nature of the enzyme, (ii) the volume of enzyme per injection, (iii) the timing and repetition of injections to be considered for future experiments. More importantly, although major variations exist between in vivo and in vitro conditions, our observations match empirical clinical evidence, providing a scientific rationale for the establishment of future clinical guidelines. The protocol could also be used in pre-clinical development to better assess gel degradation before entering animal experiments, alleviating costs and reducing the burden for animal welfare. The in vitro evaluation of filler longevity could be an interesting tool to estimate and compare the life span of pre-existing or new fillers.

## Figures and Tables

**Figure 1 molecules-28-01003-f001:**
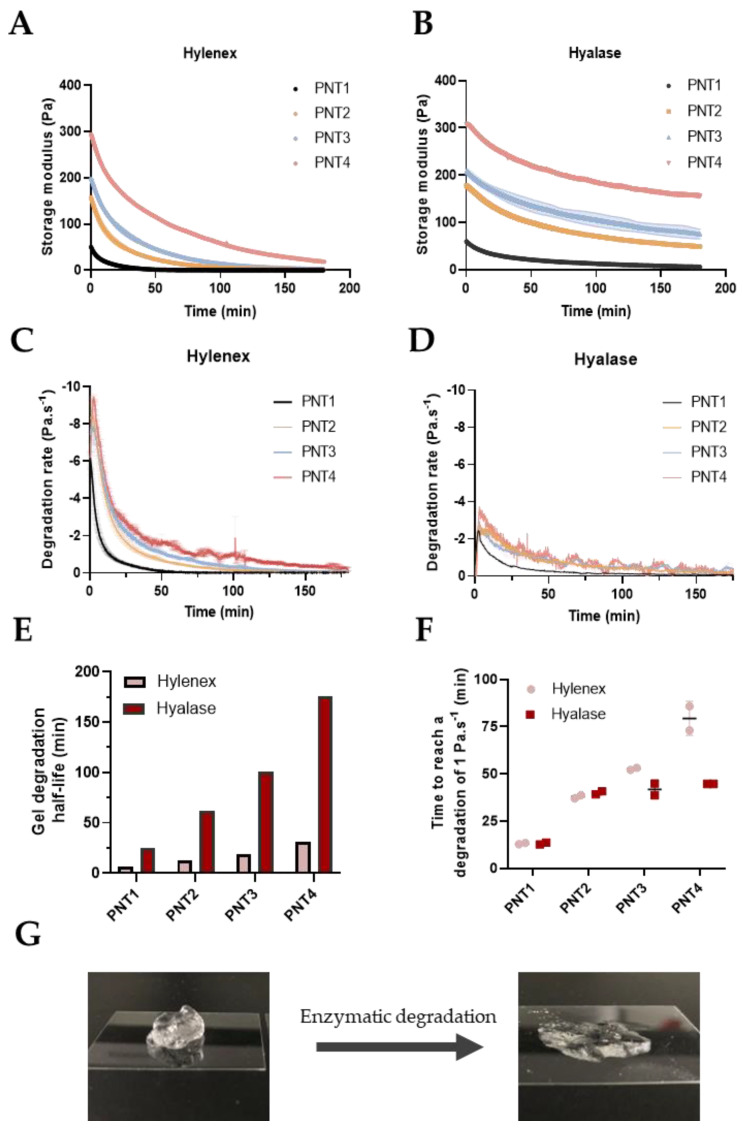
Degradation of Teosyal PNT gels in contact with Hylenex or Hyalase for 180 min, assessed by rheological time sweeps (37 °C, strain 0.1%, 1 Hz). (**A**) Degradation of PNT gels in contact with 7.1 U·mL^−1^ of Hylenex over a 3-h period, (**B**) Degradation of PNT gels in contact with 7.1 U·mL^−1^ of Hyalase over a 3-h period, degradation rates of Hylenex-treated (**C**) and Hyalase-treated (**D**) PNT gels calculated from the first derivatives, (**E**) time required to reach 50% of gel degradation (half-life) of PNT gels with Hylenex or Hyalase enzymes, (**F**) time needed to reach an enzymatic degradation speed inferior to 1 Pa·s^−1^, (**G**) example of a PNT4 gel before (left) and after (right) treatment with the Hylenex enzyme. All results are presented as mean ± S.D, n = 2.

**Figure 2 molecules-28-01003-f002:**
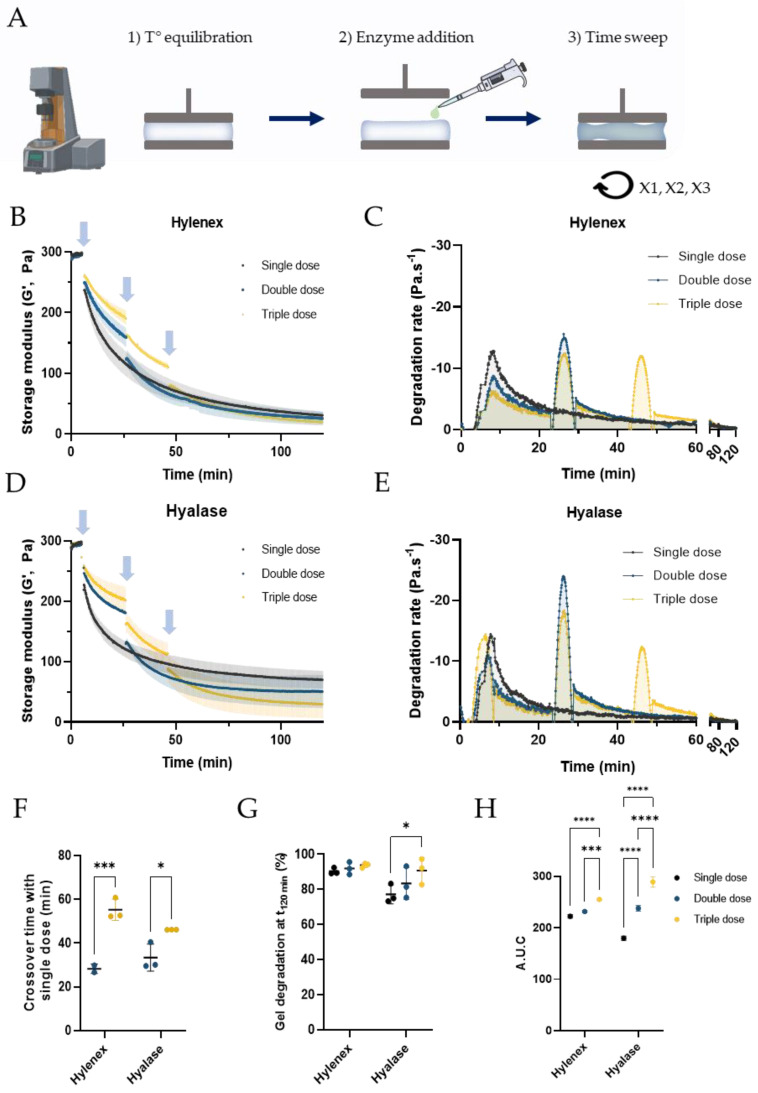
Evaluation of PNT4 degradation by single or multidose injections of Hylenex or Hyalase enzymes, followed by rheological time sweeps. (**A**) Schematic of the method used to degrade the PNT4 gels by single or multidose enzyme injections, (**B**) Evolution of the storage modulus of PNT4 gels over time, measured by time sweeps (37 °C, strain 0.1%, 1 Hz) after single, double, or triple injections of the Hylenex enzyme. For all conditions, the total volume of enzyme added is 90 μL, for a total enzymatic concentration of 34.6 U·mL^−1^; blue arrows indicate enzyme addition, (**C**) First derivative curves of single, double, and triple dose injections, (**D**) Evolution of the storage modulus of PNT4 gels over time, measured by time sweeps (37 °C, strain 0.1%, 1 Hz) after single, double, or triple injections of the Hyalase enzyme. For all conditions, the total volume of enzyme added is 60 μL, for a total number of 277.8 U·mL^−1^; blue arrows indicate enzyme addition, (**E**) First derivative curves of single, double, and triple dose injections; blue arrows indicate enzyme injections with the Hyalase enzyme, (**F**) Crossover time at which multidose storage moduli are lower than with a single dose, (**G**) Storage moduli at plateau obtained after single, double, or triple addition of enzyme, (**H**) area under the curve (AUC) extracted from the degradation rates of PNT4 gels using both enzymes. All results are presented as mean ± S.D, n = 3.

**Figure 3 molecules-28-01003-f003:**
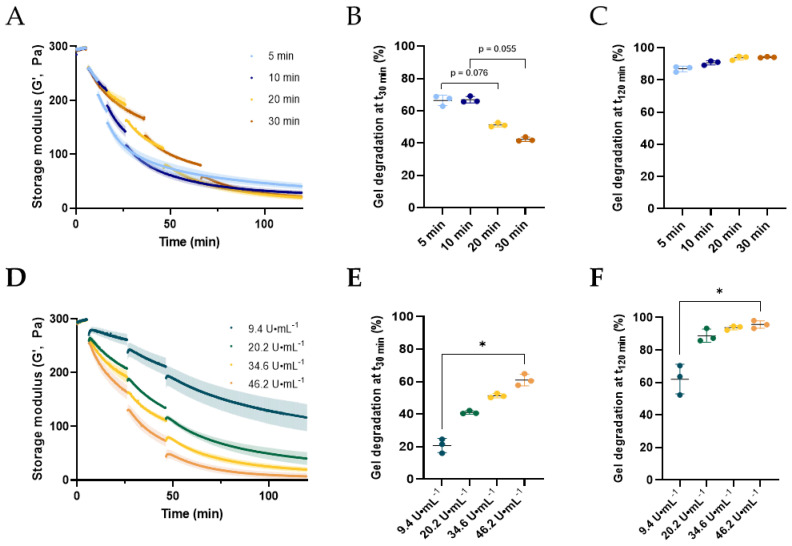
Refinement of PNT4 degradation parameters by triple injections of Hylenex, followed by rheological time sweeps. (**A**) Evolution of the storage modulus of PNT4 gels over time, measured by time sweeps (37 °C, strain 0.1%, 1 Hz), following triple injections of the Hylenex enzyme (34.6 U·mL^−1^) with injection spans of 5, 10, 20, and 30 min, Percentage of gel degradation after 30 (**B**) or 120 (**C**) min for each injection span, compared to the initial storage modulus set at 300 Pa, (**D**) Evolution of the storage modulus of PNT4 gels over time, measured by time sweeps (37 °C, strain 0.1%, 1 Hz), following triple injections of the Hylenex enzyme 20 min apart, with enzymatic unit numbers of 9.4, 20.2, 34.6, and 46.2 units of HAase per mL of gel, percentage of gel degradation after 30 (**E**) or 120 (**F**) min for each enzymatic unit number, compared to the initial storage modulus set at 300 Pa. All results are presented as mean ± S.D, n = 3. Mann–Whitney non-parametric tests were performed on (**B**,**C**,**E**,**F**), with a confidence level α = 0.05 (* *p* < 0.05).

**Figure 4 molecules-28-01003-f004:**
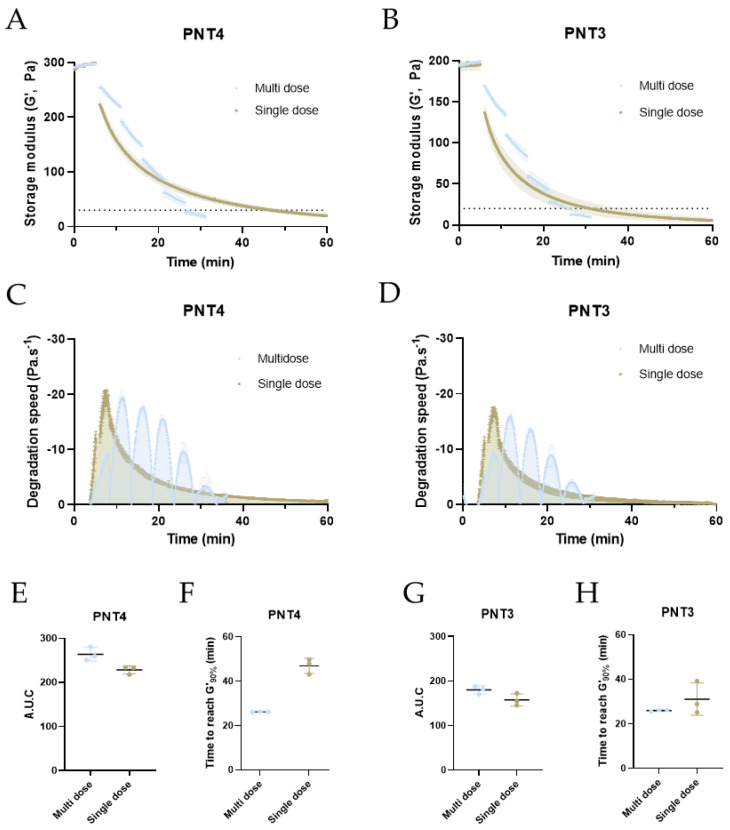
Optimization of PNT4 and PNT3 degradation by multiple injections of 38.3 U·mL^−1^ Hylenex every 5 min, followed by rheological time sweeps. Evolution of the storage modulus of PNT4 (**A**) and PNT3 (**B**) gels over time, measured by time sweeps (37 °C, strain 0.1%, 1 Hz) following single or multiple injections of the Hylenex enzyme (38.3 U·mL^−1^ per injection, 5 min injection span for multidose), total volume and enzymatic units were equivalent to single and multidose. Dotted lines represent 10% of initial G’, First derivative curves of PNT4 (**C**) and PNT3 (**D**) gels for single and multidose injections, area under the curve of PNT4 (**E**) and PNT3 (**G**) gels calculated from the first derivative curves, time required to reach 90% of gel degradation for PNT4 (**F**) and PNT3 (**H**) gels, relative to the initial storage modulus. All results are presented as mean ± S.D, n = 3.

**Table 1 molecules-28-01003-t001:** List of the TEOSYAL fillers investigated in this study.

Filler	Abbreviation	Storage Modulus (Pa) ^a^	Delta (°) ^a^	HA Concentration (mg/mL)	Degree of Modification (%) ^b^	Batch References
TEOSYAL PNT1	PNT1	58.6	19.5	15	1.9	TPRL_22142AA
TEOSYAL PNT2	PNT2	138.9	16.2	23	3.1	TP30L_220916A TP30L_22142AA TP30L_22121DA
TEOSYAL PNT3	PNT3	145.8	12.1	23	3.6	TP27L_221117A TP27L2214IA
TEOSYAL PNT4	PNT4	263.3	6.6	23	4.1	TPUL_22142CA TPUL_22142BA TPUL_221029A TPUL_22232IA TPUL_222232H

^a^ As determined by [10] at 25 °C, ^b^ measured by ^1^H NMR.

## Data Availability

Not applicable.

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
