# Peer review of "Multidose Hyaluronidase Administration as an Optimal Procedure to Degrade Resilient Hyaluronic Acid Soft Tissue Fillers"

_molecules, 2023, doi:10.3390/molecules28031003_

Round 1

Reviewer 1 Report

Dear authors, 

The study is very interesting. 

The introduction is sufficient and reflects very well the importance of the study.

The results are well described and are very important scientifically because there are no guidelines reported in the literature, regarding degradation of the fillers.

The material and methods of a study cannot be presented after results, discussions and conclusion (conclusions end at line 410 and material and methods begin at line 411)

Thank you

Author Response

We would like to kindly thank reviewer 1 for reviewing our manuscript and highlighting our willingness to give a scientific rationale for the establishment of new guidelines for the controlled degradation of HA fillers.

Following reviewer’s comment, we modified our manuscript by inserting the Material and Methods section prior to the results.

Reviewer 2 Report

Your article proposes an original protocol based on rheological measures to determine the ability of hyaluronidase to degrade HA filler. Congratulation on your scientific contribution concerning the definition of the correct timing of hyaluronidase injection for HA filler degradation. I suggested that the editor should publish your article.

Author Response

We gratefully thank reviewer 2 for reviewing our manuscript and for acknowledging the relevance of our work. We hope that our manuscript will help practitioners defining clear guidelines for the in vivo degradation of soft tissue HA-based fillers.

Reviewer 3 Report

Authors are to be commended for describing a scientific evaluation of hyaluronidase on HA filers

I would suggest only some small improvements: a short discussion on the potency of hyaluronidases, how it is measured and why in units; a comparison of the two hyaluronidases used; and finally would move "matherials and methods" as paragraph number 2 since the reading of the text is facilitated this way

Author Response

We would like to thank reviewer 3 for his review and for his thoughtful comments.

We added a short discussion on the potency of HAase and how it is measured, also emphasizing that the standards used to measure the enzymatic potency may vary between European and American guidelines:

P1, l36-40 :” Their potency, or enzymatic activity, is determined by turbidimetric assays, which may vary between European and American standards[4]. It is measured in units, defining the amount of enzyme catalyzing the conversion of a substrate per unit of time.  

In our manuscript, we mentioned that Hylenex is a recombinant human HAase while the Hyalase originates from ovine origin (P12, l328-329). In addition, the number of Units are mentioned in the Material and Methods section. This is all the information we have at our disposal. However, we also added a small paragraph to better explain that the differences in the methodology between the American and European Pharmacopeia may explain the observed differences in the kinetics of gel degradation.

P12, l332-334: “ In addition, methods for determining the enzymatic potency may vary between the American and European Pharmacopeia. A direct comparison between the two enzymatic activities is therefore deemed impossible, calling for precautions during clinical practice[30]. »

Following reviewer’s comment, we modified our manuscript by inserting the Material and Methods section prior to the results.